# GLOBAL CONVERGENCE AND STABILITY OF STOCHASTIC GRADIENT DESCENT

## ABSTRACT

In machine learning, stochastic gradient descent (SGD) is widely deployed to train models using highly non-convex objectives with equally complex noise models. Unfortunately, SGD theory often makes restrictive assumptions that fail to capture the non-convexity of real problems, and almost entirely ignore the complex noise models that exist in practice. In this work, we make substantial progress on this shortcoming. First, we establish that SGD's iterates will either globally converge to a stationary point or diverge under nearly arbitrary nonconvexity and noise models. Under a slightly more restrictive assumption on the joint behavior of the non-convexity and noise model that generalizes current assumptions in the literature, we show that the objective function cannot diverge, even if the iterates diverge. As a consequence of our results, SGD can be applied to a greater range of stochastic optimization problems with confidence about its global convergence behavior and stability.

## 1 INTRODUCTION

Stochastic Gradient Descent (SGD) is a dominant algorithm for solving stochastic optimization problems that arise in machine learning, and has expanded its reach to more complex problems from estimating Gaussian Processes (Chen et al., 2020), covariance estimation in stochastic filters (Kim et al., 2021), and systems identification (Hardt et al., 2016; Zhang & Patel, 2020). Accordingly, understanding SGD's behavior has been crucial to its reliable application in machine learning and beyond. As a result, SGD's theory has greatly advanced from a variety of perspectives: global convergence analysis (Lei et al., 2019; Gower et al., 2020; Khaled & Richtárik, 2020; Mertikopoulos et al., 2020; Patel, 2020), local convergence analysis (Mertikopoulos et al., 2020), greedy and global complexity analysis (Gower et al., 2020; Khaled & Richtárik, 2020), asymptotic weak convergence (Wang et al., 2021), and saddle point analysis (Fang et al., 2019; Mertikopoulos et al., 2020; Jin et al., 2021).

While all of these perspectives add new dimensions to our understanding of SGD, the global convergence analysis of SGD is the foundation as it dictates whether local analyses, complexity analyses or saddle point analyses are even warranted. Unfortunately, current global convergence analyses of SGD make restrictive assumptions that fail to capture actual stochastic optimization problems arising in machine learning. In particular, global convergence analyses of SGD assume:

1. a global Hölder constant for the gradient of the objective function (e.g., Reddi et al., 2016a; Ma & Klabjan, 2017; Zhou et al., 2018; Bassily et al., 2018; Lei et al., 2019; Li & Orabona, 2019; Gower et al., 2020; Khaled & Richtárik, 2020; Mertikopoulos et al., 2020; Patel, 2020; Jin et al., 2021);
2. unrealistic noise models (e.g., uniformly bounded variance) for the stochastic gradients (e.g., Reddi et al., 2016b; Ma & Klabjan, 2017; Hu et al., 2019; Bi & Gunn, 2019; Zou et al., 2019; Mertikopoulos et al., 2020).

While such assumptions often make the analysis simpler, the resulting global convergence results would not even apply to simple neural network models for binary classification (see Appendix A).

To address this gap, we analyze the global convergence of SGD on nonconvex stochastic optimization problems that capture many actual machine learning applications. In particular, we assume local Hölder continuity of the gradient function (see Assumption 2), which substantially relaxes the global Hölder assumption of all previous works. Second, we assume that the noise model of the stochastic gradients is bounded by an arbitrary upper semi-continuous function of the parameter (see Assumption 4), and can even have infinite variance (c.f., Wang et al., 2021), which generalizes the assumptions of all previous work. With these two general assumptions, we prove that the iterates of SGD will either converge to a stationary point or diverge to infinity with probability one (see Theorem 2). Owing to our result, SGD can be applied to actual empirical risk minimization problems with guarantees about its asymptotic behavior.

In the process of proving Theorem 2, we also prove another almost remarkable claim about the behavior of SGD's iterates: that of all the possible asymptotic behaviors of the iterates (e.g., convergence to a fixed point, convergence to a manifold, limit cycles, oscillation between points, divergence), even with rather arbitrary noise models, the *only two possibilities* are either the iterates converge to a fixed point or they diverge (see Theorem 1). Note, we cannot expect this outcome apriori even in such a simple context as applying SGD with fixed step sizes to solve a consistent linear system: the iterates may terminate in a cycle and, thus, fail to converge to a fixed point (Motzkin & Schoenberg, 1954, Theorem 2). Thus, from a practical perspective, applying SGD to nonconvex problems with exotic noise models, which may initially cause concern, will either result in convergence to a stationary point or divergence of the iterates.

While Theorem 2 is patently useful, SGD's iterates diverging may cause some concern, especially when optimizing regularized empirical risk functions that are guaranteed to be coercive (i.e., the objective diverges to infinity as the argument tends to infinity). To address this issue, we generalize the notion of expected smoothness (see Khaled & Richtárik, 2020) to an assumption about the joint behavior of the gradient function, the noise model, and the local Hölder constant (see Assumption 5) to prove that, regardless SGD's iterates' behavior, the objective function will remain finite and the gradient function will converge to zero with probability one (see Theorem 3). Thus, from a practical perspective, if we can apply Theorem 3 to a coercive objective function, we are guaranteed that the iterates *cannot* diverge, and, consequently, must converge to a stationary point.

Furthering the practical value of our results, as alluded to previously, our results enable the use of downstream analyses. Specifically, our results allow for SGD's iterates to converge to stationary point or diverge; as a result, when SGD's iterates converge to a stationary point, saddle point escape analyses (e.g., Fang et al., 2019; Mertikopoulos et al., 2020; Jin et al., 2021) can be applied to ensure that the stationary point is a local minimizer. Moreover, when SGD is converging to a stationary point, local convergence rate analyses can also be supplied, which can inform adaptive step size rules and stopping criteria (Patel, 2020).

Finally, from a theoretical perspective, we innovate two new analysis strategies to deal with the generality of the local Hölder continuity assumption on the gradient, and our general noise model assumption. We term these two strategies the pseudo-global strategy and the local strategy. We develop the pseudo-global strategy to prove global convergence (i.e., Theorem 2), while we develop the local strategy to prove stability (i.e., Theorem 3). We believe that both of our strategies are of independent interest to theoreticians.

**Contribution Summary.** To summarize, we study the behavior of SGD under much more realistic assumptions than what is currently in the literature; namely, we consider local Hölder continuity and general noise models. In this context, we are able to show (1) that the iterates must either converge to a fixed point or that they might diverge (Theorem 1); and (2), when the iterates converge to a fixed point, it must be a stationary point of the objective function (Theorem 2).

Moreover, under a slightly more restrictive assumption—which still generalizes current assumptions in the literature—, we show that, regardless of iterate behavior, the objective function will converge to a finite random variable (i.e., SGD is stable) and the gradient function will converge to zero (Theorem 3).

Finally, we develop two new analysis strategies, the pseudo-global strategy and the local strategy, that are of independent interest to theoreticians in machine learning and stochastic optimization.

**Organization.** The remainder of the paper is organized as follows. In Section 2, we specify the stochastic optimization problem that we will study, including a formal statement of all assumptions. In Section 3, we specify Stochastic Gradient Descent (SGD) and the properties of the needed properties of the learning rate. In Section 4, we prove and discuss our main results and highlight key steps, while leaving the rest to the appendix. In Section 5, we conclude this work.

## 2 STOCHASTIC OPTIMIZATION

We consider solving the optimization problem

$$\min_{\theta \in \mathbb{R}^p} \{F(\theta) := \mathbb{E}\left[f(\theta, X)\right]\}, \tag{1}$$

where $F$ maps $\mathbb{R}^p$ into $\mathbb{R}$; $f$ maps $\mathbb{R}^p$ and the co-domain of the random variable $X$ into $\mathbb{R}$; and $\mathbb{E}$ is the expectation operator. As we require gradients, we take $F$ and $f$ to differentiable in $\theta$, and denote its derivatives with respect to $\theta$ by $\dot{F}(\theta)$ and $\dot{f}(\theta, X)$. With this notation, we make the following general assumptions about the deterministic portion of the objective function.

**Assumption 1.** *There exists $F_{l.b.} \in \mathbb{R}$ such that $\forall \theta \in \mathbb{R}^p$, $F_{l.b.} \leq F(\theta)$.*

**Assumption 2.** *There exists $\alpha \in (0, 1]$ such that $\dot{F}(\theta)$ is locally $\alpha$-Hölder continuous.*

Assumptions 1 and 2 would even be considered mild in the context of non-convex deterministic optimization, in which it is also common to assume that the objective function well-behaved level sets (e.g., Nocedal & Wright, 2006, Theorems 3.2, 3.8, 4.5, 4.6). Importantly, Assumption 2 relaxes the common restrictive assumption of globally Hölder continuous gradient functions that is common in other analyses.

Our final step is to make some assumptions about the stochastic portion of the objective function. The first assumption requires the stochastic gradients to be unbiased, which can readily be relaxed (Bottou et al., 2018). The second assumption allows for a generic noise model for an $\alpha$-Hölder continuous gradient function, and even allows for the second moment to not exist when $\alpha < 1$ (c.f. Wang et al., 2021).

**Assumption 3.** *For all $\theta \in \mathbb{R}^p$, $\dot{F}(\theta) = \mathbb{E}[\dot{f}(\theta, X)]$.*

**Assumption 4.** *Let $\alpha \in (0, 1]$ be as in Assumption 2. There exists an upper semi-continuous function $G(\theta)$ such that $\mathbb{E}[\|\dot{f}(\theta, X)\|_2^{1+\alpha}] \leq G(\theta)$.*

> We will show that Assumptions 1 to 4 are sufficient for a global convergence result.

**Remark 1.** *It is entirely possible that $\mathbb{E}[\|\dot{f}(\theta, X)\|_2^{1+\alpha}]$ is (at least) upper semi-continuous, and to set $G(\theta)$ equal to this function. In the case that $\mathbb{E}[\|\dot{f}(\theta, X)\|_2^{1+\alpha}]$ is not upper semi-continuous, it is possible to specify $G(\theta)$ as the upper envelope of $\mathbb{E}[\|\dot{f}(\theta, X)\|_2^{1+\alpha}]$ (i.e., the its limit supremum function). However, it is unlikely that $\mathbb{E}[\|\dot{f}(\theta, X)\|_2^{1+\alpha}]$ nor its upper envelope are easy to specify explicitly, and it is more likely to be able to find an upper bound.*

In order to show that the objective function cannot diverge (i.e., to prove stability), we will need an additional assumption. This assumption will relate the gradient function, noise model and variation on the local Hölder constant. To begin, we define the variation on the local Hölder constant. Let $\alpha \in (0, 1]$ be as in Assumption 2 and $\epsilon > 0$ be arbitrary, and define

$$\mathcal{L}_\epsilon(\theta) = \begin{cases} \sup_\varphi \left\{ \frac{\|\dot{F}(\varphi) - \dot{F}(\theta)\|_2}{\|\varphi - \theta\|_2^\alpha} : \|\varphi - \theta\|_2 \leq (G(\theta) \vee \epsilon)^{\frac{1}{1+\alpha}} \right\} & \text{if this quantity is nonzero} \\ \epsilon & \text{otherwise,} \end{cases} \tag{2}$$

where $\vee$ indicates the maximum between two quantities. Note, the choice of $\epsilon$ is irrelevant, and they can be distinct for the two cases in the definition of $\mathcal{L}_\epsilon$, but we fix them to be the same for simplicity. Importantly, the quantity, $\mathcal{L}_\epsilon$, is defined at every parameter $\theta$ under Assumption 2.

With this quantity, we can state a nonintuitive, technical assumption that is needed to prove stability.

**Assumption 5.** *There exists* $C_1, C_2, C_3 \geq 0$ *such that,* $\forall \theta \in \mathbb{R}^p$,

$$\mathcal{L}_\epsilon(\theta)G(\theta) + \alpha \left( \frac{\left\| \dot{F}(\theta) \right\|_2^{1+\alpha}}{\mathcal{L}_\epsilon(\theta)} \right)^{1/\alpha} \leq C_1 + C_2(F(\theta) - F_{l.b.}) + C_3 \left\| \dot{F}(\theta) \right\|_2^2. \tag{3}$$

Assumption 5 generalizes Assumption 4.3(c) of Bottou et al. (2018), which is satisfied for a large swath of statistical models. Moreover, Assumption 5 generalizes the notion of expected smoothness (see Khaled & Richtárik, 2020, for a history of the assumption), which expanded the optimization problems covered by the theory of Bottou et al. (2018). Note, Assumption 5 is about the asymptotic properties of the stochastic optimization problem as the left hand side of the inequality in Assumption 5 can be bounded inside of any compact set. Thus, Assumption 5 covers a variety of asymptotic behaviors, such as $\exp(\|\theta\|_2^2)$, $\exp(\|\theta\|_2)$, $\|\theta\|_2^r$ for $r \in \mathbb{R}$, $\log(\|\theta\|_2+1)$, and $\log(\log(\|\theta\|_2+1)+1)$. Therefore, Assumption 5 holds for functions with a variety of different asymptotic behaviors.

---

We will show that Assumptions 1 to 5 are sufficient for a stability result.

---

Now that we have specified the nature of the stochastic optimization problem, we turn our attention to the algorithm used to solve the problem, namely, stochastic gradient descent (SGD).

## 3 STOCHASTIC GRADIENT DESCENT

SGD starts with an arbitrary initial value, $\theta_0 \in \mathbb{R}^p$, and generates a sequence of iterates $\{\theta_k : k \in \mathbb{N}\}$ according to the rule

$$\theta_{k+1} = \theta_k - M_k \dot{f}(\theta_k, X_{k+1}), \tag{4}$$

where $\{M_k : k+1 \in \mathbb{N}\} \subset \mathbb{R}^{p \times p}$; and $\{X_k : k \in \mathbb{N}\}$ are independent and identically distributed copies of $X$. Importantly, $\{M_k\}$ cannot be arbitrary, and the following properties specify a generalization of the Robbins & Monro (1951) conditions for matrix-valued learning rates (c.f. Patel, 2020).

The first condition requires a positive learning rate, and imposes symmetry to ensure the existence of real eigenvalues.

**Property 1.** $\{M_k : k+1 \in \mathbb{N}\}$ *are symmetric, positive definite matrices.*

The next two properties are a natural generalization of the Robbins-Monro conditions. Let $\alpha \in (0, 1]$ be as in Assumption 2.

**Property 2.** *Let* $\lambda_{\max}(\cdot)$ *denote the largest eigenvalue of a symmetric, positive definite matrix. Then,* $\sum_{k=0}^\infty \lambda_{\max}(M_k)^{1+\alpha} =: S < \infty$.

**Property 3.** *Let* $\lambda_{\min}(\cdot)$ *denote the smallest eigenvalue of a symmetric, positive definite matrix. Then,* $\sum_{k=0}^\infty \lambda_{\min}(M_k) = \infty$.

---

We will show that Properties 1 to 3 are sufficient for a global convergence result.

---

The final property ensures the stability of the condition number of $\{M_k\}$. Note, this property is readily satisfied for scalar learning rates satisfying the Robbins-Monro conditions.

**Property 4.** *Let $\kappa(\cdot)$ denote the ratio of the largest and smallest eigenvalues of a symmetric, positive definite matrix. Then, $\lim_{k\to\infty} \lambda_{\max}(M_k)^\alpha \kappa(M_k) = 0$.*

We will show that Properties 1 to 4 are sufficient for stability.

## 4 GLOBAL CONVERGENCE & STABILITY

With the stochastic optimization problem and with stochastic gradient descent (SGD) specified, we now turn our attention to what happens when SGD is applied to a stochastic optimization problem. The key step in the analysis of SGD on any objective function is to establish a bound between the optimality gap at $\theta_{k+1}$ with that of $\theta_k$. This step is achieved by using the local Hölder continuity of the gradient function and the fundamental theorem of calculus. Using Assumption 2, we first specify the local Hölder constant.

**Definition 1.** *For any $\theta, \varphi \in \mathbb{R}^p$, define*

$$L(\theta, \varphi) = \sup_{\psi} \left\{ \frac{\left\| \dot{F}(\psi) - \dot{F}(\theta) \right\|_2}{\|\psi - \theta\|_2^\alpha} : \|\psi - \theta\|_2 \leq \|\varphi - \theta\|_2 \right\}. \tag{5}$$

*Moreover, for any $R \geq 0$, let $L_R = L(0, \varphi)$ for $\|\varphi\|_2 = R$.*

**Remark 2.** *Note, when the gradient is locally Hölder continuous, $L_R$ is well defined for any $R \geq 0$.*

With this definition, we can now relate the optimality gap of $\theta_{k+1}$ with that of $\theta_k$ by using the following result.

**Lemma 1.** *Suppose Assumptions 1 and 2 hold. Then, for any $\theta, \varphi \in \mathbb{R}^p$,*

$$F(\varphi) - F_{l.b.} \leq F(\theta) - F_{l.b.} + \dot{F}(\theta)'(\varphi - \theta) + \frac{L(\theta, \varphi)}{1 + \alpha} \|\varphi - \theta\|_2^{1+\alpha}. \tag{6}$$

*Proof.* By Taylor's theorem,

$$F(\varphi) - F_{l.b.} = F(\theta) - F_{l.b.} + \int_0^1 \dot{F}(\theta + t(\varphi - \theta))'(\varphi - \theta)dt. \tag{7}$$

Now, add and subtract $\dot{F}(\theta)$ to $\dot{F}(\theta + t(\varphi - \theta))$ in the integral, then apply Assumption 2. We conclude,

$$F(\varphi) - F_{l.b.} \leq F(\theta) - F_{l.b.} + \dot{F}(\theta)'(\varphi - \theta) + L(\theta, \varphi) \|\varphi - \theta\|_2^{1+\alpha} \int_0^1 t^\alpha dt. \tag{8}$$

By computing the integral, the result follows. $\square$

Now, if we simply set $\varphi = \theta_{k+1}$ and $\theta = \theta_k$ in Lemma 1 and try to take expectations to manage the randomness of the stochastic gradient, we will run into the problem that $L(\theta_k, \theta_{k+1})$ and $\|\theta_{k+1} - \theta_k\|_2$ are dependent, and we cannot compute its expectation. In previous work, this technical challenge is waived away by using a global Hölder constant to upper bound $L(\theta_k, \theta_{k+1})$, which is unrealistic even for simple problems (see Appendix A).

To address this technical challenge, we innovate two new strategies for handling the dependence between $L(\theta_k, \theta_{k+1})$ and $\|\theta_{k+1} - \theta_k\|_2$. In both strategies, we follow the same general approach:

1. We begin by restricting our analysis to specific events, which will allow us to decouple $L(\theta_k, \theta_{k+1})$ and $\|\theta_{k+1} - \theta_k\|_2$.
2. With these two quantities decoupled, we will develop a recurrence relationship between the optimality gap at $\theta_{k+1}$ and that of $\theta_k$.

3. We apply this recurrence relationship with refinements of standard arguments or new ones to derive the desired property about the objective function.
4. Finally, we state the generality of the specific events on which we have studied SGD's iterates.

Thus, it follows, we will define two distinct series of events for the two strategies. The first strategy, which we refer to as the pseudo-global strategy, will provide the global convergence analysis. The second strategy, which we refer to as the local strategy, will provide the stability result.

## 4.1 PSEUDO-GLOBAL STRATEGY AND GLOBAL CONVERGENCE ANALYSIS

For the first strategy, which supplies the global convergence result, we study SGD on the events

$$\mathcal{B}_k(R) := \bigcap_{j=0}^{k} \left\{ \|\theta_j\|_2 \leq R \right\}, \ k+1 \in \mathbb{N}, \tag{9}$$

for every $R \geq 0$. We now try to control the optimality gap at iteration $k+1$ with that of iteration $k$, which will result in two cases.

1. (Case 1) $\mathcal{B}_{k+1}(R)$ holds we can bound $L(\theta_k, \theta_{k+1})$ by $L_R$, and $G(\theta)$ is also bounded in the ball of radius $R$ about the origin (which follows from $G$ being upper semi-continuous in Assumption 4). As a result, we could then proceed with the analysis in a manner that is similar to having a global Hölder constant.
2. (Case 2) $\|\theta_{k+1}\|_2 > R$ and $\mathcal{B}_k(R)$ holds. In this case, controlling $L(\theta_k, \theta_{k+1})$ is very challenging and, to our knowledge, was not solved before our work.

Our approach for controlling the optimality gap in both cases is supplied in the next lemma, whose proof is in Appendix C.

**Lemma 2.** *Let* $\{M_k\}$ *satisfy* Property 1. *Suppose* Assumptions 1 *to* 4 *hold. Let* $\{\theta_k\}$ *satisfy* (4). *Then,* $\forall R \geq 0$,

$$\mathbb{E}\left[ \left[ F(\theta_{k+1}) - F_{l.b.} \right] \mathbf{1} \left[ \mathcal{B}_{k+1}(R) \right] \middle| \mathcal{F}_k \right] \leq \left[ F(\theta_k) - F_{l.b.} \right] \mathbf{1} \left[ \mathcal{B}_k(R) \right]$$
$$- \lambda_{\min}(M_k) \left\| \dot{F}(\theta_k) \right\|_2^2 \mathbf{1} \left[ \mathcal{B}_k(R) \right] + \frac{L_{R+1} + \partial F_R}{1+\alpha} \lambda_{\max}(M_k)^{1+\alpha} G_R, \tag{10}$$

*where* $G_R = \sup_{\theta \in \overline{B(R)}} G(\theta) < \infty$ *with* $G(\theta)$; *and* $\partial F_R = \sup_{\theta \in \overline{B(R)}} \|\dot{F}(\theta)\|_2 (1+\alpha) < \infty$.

With this recursion and standard martingale results (Robbins & Siegmund, 1971; Neveu & Speed, 1975, Exercise II.4), the limit of $[F(\theta_k) - F_{l.b.}]\mathbf{1}[\mathcal{B}_k(R)]$ exists with probability one and is finite for every $R \geq 0$. As a result, the limit of $F(\theta_k) - F_{l.b.}$ exists and is finite on the event $\{\sup_k \|\theta_k\|_2 < \infty\}$ (see Corollary 1).

We can also use Lemma 2 to make a statement about the gradient. Specifically, we can show that the limit infimum of $\mathbb{E}[\|\dot{F}(\theta_k)\|_2^2 \mathbf{1}[\mathcal{B}_k(R)]]$ must be zero, which is now a standard argument that mimics Zoutendijk's theorem (Nocedal & Wright, 2006, Theorem 3.2). By Markov's inequality, this result implies that $\|\dot{F}(\theta_k)\|_2 \mathbf{1}[\mathcal{B}_k(R)]$ gets arbitrarily close to $0$ infinitely often (see Lemma 8). To show convergence to zero, however, is not standard. Several strategies have been developed, namely those of Li & Orabona (2019); Lei et al. (2019); Mertikopoulos et al. (2020); Patel (2020). Unfortunately, the approaches of Li & Orabona (2019); Lei et al. (2019) rely intimately on the existence of a global Hölder constant, while that of Mertikopoulos et al. (2020) requires even more restrictive assumptions. Fortunately, the approach of Patel (2020) can be improved and generalized to the current context (see Lemma 9). Thus, we show that $\lim_{k \to \infty} \|\dot{F}(\theta_k)\|_2 = 0$ on $\{\sup_k \|\theta_k\|_2 < \infty\}$ (see Corollary 2).

Our final step is to clarify the role of $\{\sup_k \|\theta_k\|_2 < \infty\}$ in the asymptotic behavior of SGD's iterates. At first glance, this event seems to imply that the iterates converge to a point. However, owing to the general nature of the noise, it is also possible, say, that the iterates approach a limit cycle or oscillate between points with the same norm. Even beyond this event, the generality of the

noise model may allow for substantial excursions between $F_{l.b.}$ and infinity (c.f., a simple random walk, which has a limit supremum of infinity and a limit infinimum of negative infinity). Thankfully, we can prove that either the iterates converge to a point or they must diverge—a result that we refer to as the Capture Theorem (see Appendix C).

**Theorem 1** (Capture Theorem). *Let $\{\theta_k\}$ be defined as in (4), and let $\{M_k\}$ satisfy Properties 1 and 2. If Assumption 4 holds, then either $\{\lim_{k\to\infty} \theta_k \text{ exists}\}$ or $\{\liminf_{k\to\infty} \|\theta_k\|_2 = \infty\}$ must occur.*

By putting together the above arguments and results, we can conclude that either SGD's iterates diverge or SGD's iterates converge to a stationary point. This is formally stated in the following theorem. See Section 1 for a discussion of the practical value of this result.

**Theorem 2** (Global Convergence). *Let $\theta_0$ be arbitrary, and let $\{\theta_k : k \in \mathbb{N}\}$ be defined according to (4) with $\{M_k : k+1\}$ satisfying Properties 1 to 3. Suppose Assumptions 1 to 4 hold. Let $\mathcal{A}_1 = \{\liminf_{k\to\infty} \|\theta_k\|_2 = \infty\}$ and $\mathcal{A}_2 = \{\lim_{k\to\infty} \theta_k \text{ exists}\}$. Then, the following statements hold.*

1. $\mathbb{P}[\mathcal{A}_1] + \mathbb{P}[\mathcal{A}_2] = 1$.
2. *On $\mathcal{A}_2$, there exists a finite random variable, $F_{\lim}$, such that $\lim_{k\to\infty} F(\theta_k) = F_{\lim}$ and $\lim_{k\to\infty} \dot{F}(\theta_k) = 0$ with probability one.*

*Proof.* By Theorem 1, we have that $\mathbb{P}[\mathcal{A}_1] + \mathbb{P}[\mathcal{A}_2] = 1$. Then, on $\mathcal{A}_2$, Corollaries 1 and 2 imply that $F(\theta_k) \to F_{\lim}$, which is finite, and $\dot{F}(\theta_k) \to 0$. □

### 4.2 LOCAL STRATEGY AND STABILITY ANALYSIS

While Theorem 2 provides a complete global convergence result, it allows for the possibility of diverging iterates. The possibility of divergent iterates raises the spectre of whether the objective function can also diverge along this sequence. That is, there is a possibility that SGD may be unstable, which would be highly unexpected and undesirable, especially when the objective function is coercive (e.g., has an $\ell^1$ penalty on the parameters).

To formalize this concept, we define a relevant notion of stability.

**Definition 2.** *Stochastic Gradient Descent is stable if*

$$\mathbb{P}\left[\limsup_{k\to\infty} F(\theta_k) = \infty\right] = 0, \tag{11}$$

*where $\{\theta_k\}$ satisfy (4).*

We now state the events that we will use to decouple the relationship between $L(\theta_k, \theta_{k+1})$ and $\|\theta_{k+1} - \theta_k\|_2$. Unlike in our pseudo-global strategy, we will make use of two sequences of events that are closely related. To define these sequences, we will first need to define stopping times. For every $j+1 \in \mathbb{N}$, define

$$\tau_j = \min\left\{k : \begin{array}{c} F(\theta_{k+1}) - F_{l.b.} > F(\theta_k) - F_{l.b.} + \dot{F}(\theta_k)'(\theta_{k+1} - \theta_k) \\ + \dfrac{\mathcal{L}_\epsilon(\theta_k)}{1+\alpha}\|\theta_{k+1} - \theta_k\|_2^{1+\alpha}, \text{ and } k > j \end{array}\right\}. \tag{12}$$

Analogously, for every $j+1 \in \mathbb{N}$, define

$$\nu_j = \min\{k : L(\theta_k, \theta_{k+1}) > \mathcal{L}_\epsilon(\theta_k), \text{ and } k > j\}. \tag{13}$$

Now, we will use (12) to establish the stability of the objective function, and we will use (13) to show that the gradient function must tend to zero. Just as we did with $\mathcal{B}_k(R)$, we will derive a recursion on the optimality gap over the events $\{\{\tau_j > k\} : k+1 \in \mathbb{N}\}$. Of course, just as before, the main challenge in deriving a recursive formula is to address $\{\tau_j = k\}$. Our solution is supplied in the following lemma, whose proof is in Appendix D.

**Lemma 3.** *Let $\{M_k\}$ satisfy Property 1. Suppose Assumptions 1 to 4 hold. Let $\{\theta_k\}$ satisfy (4). Then, for any $j + 1 \in \mathbb{N}$ and $k > j$,*

$$\mathbb{E}\left[ (F(\theta_{k+1}) - F_{l.b.})\, \boldsymbol{I}\left[ \tau_j > k \right] \middle| \mathcal{F}_k \right] \leq \left( F(\theta_k) - F_{l.b.} - \dot{F}(\theta_k)' M_k \dot{F}(\theta_k) \right) \boldsymbol{I}\left[ \tau_j > k - 1 \right]$$

$$+ \frac{\lambda_{\max}(M_k)^{1+\alpha}}{1+\alpha} \left[ \mathcal{L}_\epsilon(\theta_k) G(\theta_k) + \alpha \left[ \frac{\left\| \dot{F}(\theta_k) \right\|_2^{1+\alpha}}{\mathcal{L}_\epsilon(\theta_k)} \right]^{1/\alpha} \right] \boldsymbol{I}\left[ \tau_j > k - 1 \right]. \tag{14}$$

From Lemma 3, there is a clear motivation for Assumption 5. Indeed, if we apply Assumption 5, Lemma 3 produces the following simple recursive relationship.

**Lemma 4.** *If Assumptions 1 to 5, and Properties 1 and 4 hold, and $\{\theta_k\}$ satisfy (4), then there exists a $K \in \mathbb{N}$ such that for any $j + 1 \in \mathbb{N}$ and any $k \geq \min\{K, j + 1\}$,*

$$\mathbb{E}\left[ (F(\theta_{k+1}) - F_{l.b.}) \boldsymbol{I}\left[ \tau_j > k \right] \middle| \mathcal{F}_k \right]$$

$$\leq \left( 1 + \lambda_{\max}(M_k)^{1+\alpha} \frac{C_2}{1+\alpha} \right) (F(\theta_k) - F_{l.b.}) \boldsymbol{I}\left[ \tau_j > k - 1 \right] \tag{15}$$

$$- \frac{1}{2} \lambda_{\min}(M_k) \left\| \dot{F}(\theta_k) \right\|_2^2 \boldsymbol{I}\left[ \tau_j > k - 1 \right] + \lambda_{\max}(M_k)^{1+\alpha} \frac{C_1}{1+\alpha}.$$

Just as in the pseudo-global strategy, Lemma 4 can be combined with standard martingale results (Robbins & Siegmund, 1971; Neveu & Speed, 1975, Exercise II.4) to conclude that the limit of $F(\theta_k)$ exists and is finite on the event $\cup_{j=0}^\infty \{\tau_j = \infty\}$ (see Corollary 3). Also as in the pseudo-global strategy, by improving on the arguments in Patel (2020), we show that $\lim_k \dot{F}(\theta_k) = 0$ on the event $\cup_{j=0}^\infty \{\nu_j = \infty\}$ (see Corollary 4).

Finally, we show that $\cup_{j=0}^\infty \{\tau_j = \infty\}$ and $\cup_{j=0}^\infty \{\nu_j = \infty\}$ are probability one events (see Theorem 5). In other words, we show that eventually $\mathcal{L}_\epsilon(\theta_k)$ will always dominate $L(\theta_k, \theta_{k+1})$. This statement should not come as a surprise on the event $\{\lim_k \theta_k \text{ exists}\}$, but is slightly surprising that it must also hold on $\{\lim_k \|\theta_k\|_2 = \infty\}$. By combining these results, we can conclude as follows.

**Theorem 3** (Stability). *Let $\theta_0$ be arbitrary, and let $\{\theta_k : k \in \mathbb{N}\}$ be defined according to (4) with $\{M_k : k + 1\}$ satisfying Properties 1 to 3. Suppose Assumptions 1 to 5 hold. Then,*

1. *There exists a finite random variable, $F_{\lim}$, such that $\lim_{k \to \infty} F(\theta_k) = F_{\lim}$ with probability one;*
2. $\lim_{k \to \infty} \dot{F}(\theta_k) = 0$ *with probability one.*

*Proof.* Using Corollary 3, we conclude that $\exists F_{\lim}$ that is finite such that $\lim_k F(\theta_k) = F_{\lim}$ on $\cup_{j=0}^\infty \{\tau_j = \infty\}$. Using Corollary 4, we conclude that $\lim_k \dot{F}(\theta_k) = 0$ on $\cup_{j=0}^\infty \{\nu_j = \infty\}$. Finally, we apply Theorem 5 to conclude that $\mathbb{P}[\cup_{j=0}^\infty \{\nu_j = \infty\}] = \mathbb{P}[\cup_{j=0}^\infty \{\tau_j = \infty\}] = 1$. $\qquad\square$

See Section 1 for a brief discussion of the practical consequences of Theorem 3.

## 5 CONCLUSION

In this work, we further filled the gap between SGD's use in practice and SGD's theory; that is, SGD is often applied in situations where the non-convexity of the objective and noise model are *not* covered by existing theory. Thus, we focused on analyzing SGD in a context that was more realistic to machine learning problems in two ways. First, we eliminated the unrealistic assumption that the gradient of the objective function is globally Hölder continuous, and replaced it with a local Hölder continuity assumption. Second, we allowed for arbitrary noise models. This latter innovation suggests the possibility of potentially undesirable outcomes: the iterates can enter a limit cycle or they can oscillate. Perhaps most interestingly, in the process of establishing our global convergence

result, we showed that these undesirable outcomes were impossible (Theorem 1): either the iterates converge to a stationary point or they diverge (Theorem 2).

The possibility of the iterates diverging also raises the question of what happens to the objective function when the iterates diverge. Thus, under an additional assumption on the joint behavior of the local Hölder constant, noise model and gradient function, we showed that the objective function remains finite along any iterate sequence (i.e., SGD is stable), and the gradient function converges to zero along this iterate sequence (Theorem 3). Surprisingly, in the process of proving this result, we showed that, eventually, the constant $L(\theta_k, \theta_{k+1})$ is eventually well controlled by $\mathcal{L}_\epsilon(\theta_k)$, *even if the iterate sequence is diverging*.

These results have several practical consequences, which we enumerate presently. First, by establishing the global convergence and stability of SGD over nonconvex functions with general noise models, we enable all other types of analyses: local convergence rate analyses, local weak converge analyses, saddle point escape time analyses, and complexity analyses. Second, our global convergence and stability results can be applied to realistic machine learning optimization problems, and provide guarantees about whether SGD will find a stationary point. Finally, as SGD continues to be applied to more complex problems outside of machine learning, our results are able to supply confidence in the performance of the algorithm on such problems.

From a theoretical perspective, we innovated two new analysis strategies and substantially refined a number of argument approaches currently in the literature. We anticipate that these new argument strategies will be of substantial interest to those working in machine learning and stochastic optimization.

One of the important issues not considered in this work is the nature of $\mathcal{L}_\epsilon(\theta)$. In particular, there are functions that may admit upper bounds on $\mathcal{L}_\epsilon(\theta)$ that ensure Assumption 5 is satisfied, yet using actual value of $\mathcal{L}_\epsilon(\theta)$ would suggest that Assumption 5 is not satisfied. Exploring this issue will be the subject of future work. Another area of future effort will be to construct a realistic example in which Assumption 5 is not satisfied, and to show that either (1) the iterates diverge, which implies that Assumption 5 is sufficient and may be necessary; or (2) the iterates converge, which implies that Assumption 5 is sufficient but not necessary.

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
