# OpenReview forum: "Global Convergence and Stability of Stochastic Gradient Descent"
_ICLR.cc/2022/Conference — ICLR 2022 Submitted_

### Official Review · Reviewer_tECd · 2021-10-24

**Correctness:** 4
**Technical Novelty And Significance:** 3
**Empirical Novelty And Significance:** 3
**Recommendation:** 6
**Confidence:** 4

**Details Of Ethics Concerns:**

There are no ethics concerns.

**Main Review:**

The paper is clearly written and the results seem to be interesting. Here are some comments/suggestions:

- Assumption 5 is a bit complicated and difficult to understand. It is not quite clear how this assumption comes. I would suggest the authors to give some discussions on its motivation and its connection to the existing assumptions in the literature.

- The results in the paper are of asymptotical nature, i.e., related to the limiting behavior of SGD if the iteration number goes to infinity. In practice, we are perhaps more interested in the nonasymptotical behavior, i.e., how the convergence rate is w.r.t. the iteration number. Is it possible to provide some nonasymptotical results under the local Holder continuity  assumption and the general noise model assumption? If not, are there any essential difficulties.

- The authors consider general stepsize in the form of matrices. While this stepsize is very general, in practice the most used stepsize the scalar stepsizes. If we consider scalar stepsizes, can the global convergence and stability results be further improved?

- Note that $L_R$ is defined as $L(0,\psi)$ for $\|\psi\|_2=R$. However, there are many $\psi$ with $\|\psi\|_2=R$. Can these different $\psi$ lead to the same $L(0,\psi)$. If not, then $L_R$ would not be well defined.

Typos:

page 2: the the

page 4: holds for functions a variety

page 6: our for

eq (9): Does $\\{\|\theta_j\|_2\leq R\\}$ mean $\\{X:\|\theta_j\|_2\leq R\\}$?

page 6: be get arbitrarily close

**Summary Of The Paper:**

In this paper, the authors study the behavior of SGD under very general assumptions. The existing global convergence of SGD requires two restrictive assumptions: a global Holder continuity for gradients and unrealistic noise models for stochastic gradients. This paper relaxes the global Holder continuity assumption to a local Holder continuity assumption, and consider general noise model assumptions. The main results include global convergence and stability. By global convergence, the authors show that either the iterates converge to a stationary point or they diverge. By stability, the authors show that the objective function remains finite along any iterate sequence. In the deduction, the authors introduce novel techniques to decouple some dependencies encountered in considering general assumptions.

**Summary Of The Review:**

The convergence and stability analysis of SGD in the general assumption seems interesting. The authors also introduce some techniques in the analysis.

---

> ### Author Response · Authors · 2021-11-20
> **Response to Reviewer tECd**
>
> Thank you for the detailed feedback and the grammatical corrections.
>
> 1. We have expanded the discussion of Assumption 5. See page 4.
>
> 2. These techniques can be used to prove non-asymptotic results. In our opinion, such results are greedy/optimistic in the nonconvex case as they tend to say nothing about a given realization of SGD, nor can they be operationalized without great expense.
>
> 3. We could remove Property 4. Beyond this, we cannot say more.
>
> 4. By the definition, $L_R = L(0, \psi)$ for any $|\psi|=R$. The definition is consistent.
>
> Typos: 1, 2, 3, and 5 have been fixed. Thank you!
> Typo: 4. It is not clear how $X$ is being used here so we will likely make a statement that is known to the reviewer. Implicitly, $X,X_1,X_2,\ldots$ are defined on a (complete) probability space $(\Omega, \mathcal{F}, \mathbb{P})$. Hence, $\theta_j$ is a function of $X_1(\omega), X_2(\omega),\ldots,X_j(\omega)$ for $\omega \in \Omega$. So we can write, $\theta_j = \theta_j(\omega)$. So, $\lbrace |\theta_j|_2 \leq R \rbrace = \lbrace \omega \in \Omega : | \theta_j(\omega)|_2 \leq R \rbrace$. This is generally understood.

---

> > ### Comment · Reviewer_tECd · 2021-11-29
> > **Thank you for your response.**
> >
> > My concerns are well addressed.

---

### Official Review · Reviewer_TjZW · 2021-10-31

**Correctness:** 4
**Technical Novelty And Significance:** 3
**Empirical Novelty And Significance:** Not applicable
**Recommendation:** 6
**Confidence:** 3

**Main Review:**

Strength:
1. Analyzing the global convergence of SGD is a very important research problem in the area of machine learning and optimization.
2. The paper is well written and very good to follow.
3. The analysis strategies may be beneficial for the community.

Weakness:
Compared with previous results, the main difference is the weaker assumptions, i.e.
    (1) global Holder continuity (literature) -> local Holder continuity (this work)
    (2) bounded variance of stochastic gradient (literature) -> the stochastic gradient is upper bounded by some upper semi-continuous function (this work).
The author claims that their setting is more realistic. However, they don't provide any example for supporting this claim (especially for the second one).



**Summary Of The Paper:**

In this paper, the author studies the global convergence of SGD. Compared with previous results, the authors show that under weaker conditions, the iterates of SGD will either converge to a stationary point of the objective function or diverge.

**Summary Of The Review:**

I found the manuscript to be clearly written and technically sound. Although it has some weakness, I think it still worth a publication.

---

> ### Author Response · Authors · 2021-11-20
> **See Appendix A.2**
>
> Thank you for the comments. Appendix A.2 has a simple, realistic counter example that supports both claims. The details were also worked out in the original submission.

---

### Official Review · Reviewer_bDLV · 2021-11-01

**Correctness:** 4
**Technical Novelty And Significance:** 3
**Empirical Novelty And Significance:** Not applicable
**Recommendation:** 6
**Confidence:** 3

**Main Review:**

The strength of the paper is that it studies the global convergence and stability of SGD under a fairly general non-convex setting. It is nice that unrealistic uniform boundedness noise assumption often used in the literature can be removed. One weakness is that some of the assumptions is less intuitive, and needs more explanations, discussions and justifications, especially Assumption 5. Also, the authors should perhaps provide some discussions in the contributions section what are the main technical difficulties and challenges that they overcome in order to achieve these results, and whether there is any technical novelty here.

**Summary Of The Paper:**

In this paper, the authors consider the global convergence and stability of stochastic gradient descent in a fairly general non-convex setting. They are able to remove the often-assumed unrealistic uniform bounded assumption on the noise, and also relax the global Holder assumption in the literature. Their discussions in Appendix A provide an example for which the uniform bounded assumption on the noise commonly assumed in the literature fails. Their global convergence says that under some relatively weak assumptions, SGD either diverges or the objective converges to a finite random variable and the gradient converges to zero. This excludes the bad outcomes, e.g. limit cycle or oscillation. Their stability result says that under a stronger assumption, SGD's objective converges to a finite random variable and the gradient converges to zero with probability one.

**Summary Of The Review:**

(1) I'm a little bit confused with Assumption 4. Would it be enough to simply assume that $\mathbb{E}\Vert\dot{f}(\theta,X)\Vert_{2}^{1+\alpha}$ is finite for any $\theta$? Because it seems that if so, one can always find a function $G(\theta)$ to upper bound this? Or your assumption is stronger than it seems? I understand that $G(\theta)$ appears in (2) and Assumption 5. But it is not clear to me why you need this in Assumption 4, and am also wondering in mathematics, is there some name for the smallest upper semi-continuous function that upper bounds? Another comment I have is that in Assumption 4, you already taken the expectation. So my guess is that most likely, $\mathbb{E}\Vert\dot{f}(\theta,X)\Vert_{2}^{1+\alpha}$ is not just upper-semi-continuous but actually continuous. Do you know any commonly used example where this is not true? If so, why don't you assume this is upper-semi-continuous and use this function instead of $G(\theta)$ in later discussions?

(2) Some discussions on the intuitions behind Assumption 5, and connecting it to some assumptions that the readers are more familiar with would be very helpful. For example, the authors mentioned that Assumption 5 is satisfied for $\theta\in\mathbb{R}$ for $\exp(\theta^{2})$, $\exp(\theta)$, $\theta^{r}$ etc. But those examples are all for one-dimensional $\theta$. How about in high-dimensions?

(3) I don't understand why the assumptions in Section 3 are called Property 1-Property 4 instead of say Assumption 1'-Assumption 4' or Assumption 6-Assumption 9. They seem to be assumptions instead of properties.

(4) In Theorem 3 (and maybe in Theorem 2 as well), can you say something about the properties of $F_{\lim}$? For example, is it $L_{p}$ for some $p$?

(5) In Theorem 3, when you have stability, can your proof techniques also provide some non-asymptotic convergence guarantees instead of just some asymptotic results? That can strengthen the paper by a lot.

(6) In the proof of Theorem 2, it's better to write "imply" instead of "supply" and in the first line of Section 4.2., maybe you can write "provides" instead of "supplies".

---

> ### Author Response · Authors · 2021-11-20
> **Response to Reviewer bDLV**
>
> Thank you for the detailed feedback.
>
> Main Review: We have expanded the discussion around Assumption 5, see page 4. We have mentioned the technical improvements in the contribution summary, and strengthened the discussion of the challenge and our strategy at the bottom of page 5.
>
> (1) A counter example is a function $\frac{1}{|\theta|}$ for $\theta \in \mathbb{R}\setminus\lbrace 0 \rbrace$ and $0$ otherwise. So assumption 4 is quite general. The other points are integrated into Remark 1 on pg 3. As an example of upper semicontinuity: find the peak of a neuron subject to activation. Above the activation threshold, there is no noise. Below the activation threshold, there is plenty of noise. This switching behavior requires upper semi-continuity.
>
> (2) The examples have been updated. See pg 4.
>
> (3) SGD's learning rate is within the control of the optimizer, so they need not be assumed. So we distinguish these as properties rather than assumptions.
>
> (4) Unfortunately, no.
>
> (5) These techniques can be used to prove non-asymptotic results. In our opinion, such results are greedy/optimistic in the nonconvex case as they tend to say nothing about a given realization of SGD, nor can they be operationalized without great expense.
>
> (6) These changes are made.

---

> > ### Comment · Reviewer_bDLV · 2021-11-29
> > **response to authors response**
> >
> > Thanks for your response.

---

### Decision · Program_Chairs · 2022-01-20

**Decision:**

Reject

**Comment:**

The paper considers the global convergence and stability of SGD for non-convex setting. The main contribution of the work seems to be to remove uniform bounded assumption on the noise, and to relax the global Holder assumption typically made. Their discussions in Appendix A provide an example for which the uniform bounded assumption on the noise commonly assumed in the literature fails.  The authors establish that SGD’s iterates will either globally converge to a stationary point or diverge  and hence tehir result exclude limit cycle or oscillation. Under a more restrictive assumption on the joint behavior of the non-convexity and noise model they also show that the objective function cannot diverge, even if the iterates diverge.

The reviewers are on the fence with this paper. While they agree that the paper is interesting, they only give it a score of weak accept (subsequent to rebuttal as well). One of the qualms is that while the authors claim the result helps show success of SGD in more natural non-convex problems, they don’t provide realistic examples supporting their claim. Further, while the extension to holder smoothness assumption while is indeed interesting, unless practical significance is shown via examples, the result is not that exciting.

From my point of view and reading, while the reviews are not extensive, i do not disagree with reviewers sentiment. Technically the paper is strong but there is a unanimous lack of strong excitement for the paper amongst reviewers. While there is this lack of more enthusiasm, given the number of strong submissions this year, I am tending towards a reject.